

# Native to designed: microbial α-amylases for industrial applications

Si Jie Lim[1,2] and Siti Nurbaya Oslan[1,2,3]

[1] Enzyme Technology Laboratory, VacBio 5, Institute of Bioscience, Universiti Putra Malaysia, Serdang, Selangor, Malaysia
[2] Enzyme and Microbial Technology (EMTech) Research Centre, Faculty of Biotechnology and Biomolecular Sciences, Universiti Putra Malaysia, Serdang, Selangor, Malaysia
[3] Department of Biochemistry, Faculty of Biotechnology and Biomolecular Sciences, Universiti Putra Malaysia, Serdang, Selangor, Malaysia

Corresponding author
Siti Nurbaya Oslan,
snurbayaoslan@upm.edu.my

## ABSTRACT

**Background.** α-amylases catalyze the endo-hydrolysis of α-1,4-D-glycosidic bonds in starch into smaller moieties. While industrial processes are usually performed at harsh conditions, α-amylases from mainly the bacteria, fungi and yeasts are preferred for their stabilities (thermal, pH and oxidative) and specificities (substrate and product). Microbial α-amylases can be purified and characterized for industrial applications. While exploring novel enzymes with these properties in the nature is time-costly, the advancements in protein engineering techniques including rational design, directed evolution and others have privileged their modifications to exhibit industrially ideal traits. However, the commentary on the strategies and preferably mutated residues are lacking, hindering the design of new mutants especially for enhanced substrate specificity and oxidative stability. Thus, our review ensures wider accessibility of the previously reported experimental findings to facilitate the future engineering work.

**Survey methodology and objectives.** A traditional review approach was taken to focus on the engineering of microbial α-amylases to enhance industrially favoured characteristics. The action mechanisms of α- and β-amylases were compared to avoid any bias in the research background. This review aimed to discuss the advances in modifying microbial α-amylases via protein engineering to achieve longer half-life in high temperature, improved resistance (acidic, alkaline and oxidative) and enhanced specificities (substrate and product). Captivating results were discussed in depth, including the extended half-life at 100 °C, pH 3.5 and 10, 1.8 M hydrogen peroxide as well as enhanced substrate (65.3%) and product (42.4%) specificities. These shed light to the future microbial α-amylase engineering in achieving paramount biochemical traits ameliorations to apt in the industries.

**Conclusions.** Microbial α-amylases can be tailored for specific industrial applications through protein engineering (rational design and directed evolution). While the critical mutation points are dependent on respective enzymes, formation of disulfide bridge between cysteine residues after mutations is crucial for elevated thermostability. Amino acids conversion to basic residues was reported for enhanced acidic resistance while hydrophobic interaction resulted from mutated hydrophobic residues in carbohydrate-binding module or surface-binding sites is pivotal for improved substrate specificity. Substitution of oxidation-prone methionine residues with non-polar residues increases the enzyme oxidative stability. Hence, this review provides conceptual advances for the future microbial α-amylases designs to exhibit industrially significant characteristics.

> However, more attention is needed to enhance substrate specificity and oxidative stability since they are least reported.

## INTRODUCTION

In recent years, protein engineering has been equipped as a powerful tool to elucidate the structural functions of proteins (enzymes) and modify them for enhanced properties to solve various global issues. In treating multidrug-resistant *Staphylococcus aureus* (MRSA) infections through lysis, the fusion of peptidoglycan hydrolase (PGHase) with an albumin binding domain (ABD) had rendered the mutant (enzymbiotic) to exhibit an elevated half-life in human blood serum, strengthening its therapeutics potential (*Sobieraj et al., 2020*). While β-glucosidase had been utilized to reduce the fossil-based fuels dependency through lignocellulosic biomass conversion into second-generation biofuels (*Contreras et al., 2020*), its engineering had improved its glucose tolerance (end-product inhibition) besides enhancing its half-life at intermediate temperature (50 °C) (*Cao et al., 2020*).

Despite the medical field and biofuel industry which are beneficial from the engineering of PGHase and β-glucosidase, respectively, α-amylases (E.C. 3.2.1.1) which catalyze the endohydrolysis of α-1,4-glycosidic linkages in starch to produce small carbohydrate moieties have been applied in various industries encompassing food and fruits processing, textile and paper, detergent, biofuel and animal feeds industries. Being a carbohydrate-active enzyme, α-amylase sequences have also been deposited and classified in the Carbohydrate- Active Enzyme (CAZy) database (http://www.cazy.org/) (*Lombard et al., 2014*). Out of 170 glycoside hydrolase (GH) families, the biggest group of α-amylases is in GH 13 with 109,801 protein sequences deposited as of March 2021.

Since α-amylases are highly demanded in various industries, their bulk productions are therefore, of huge interest in the research discipline. In addition, most industrial processes are performed at non-physiological conditions encompassing elevated temperature, extreme pH, high salinity, organic solvents, and surfactants, where some usages require specific substrates or products generated (*Nigam, 2013*; *Sudan et al., 2018*). Microbial α-amylases with desirable properties are preferable since they can be natively isolated from wild type host, heterologously expressed in the recombinant host or engineered for the desired traits (*Nigam, 2013*; *Lim, Oslan & Oslan, 2020*).

Despite the long process of isolating native α-amylases with desired traits from environmental sources, researchers have developed strategies in generating new α-amylases which suit into industrial applications through modifications of certain regions / residues in the protein sequences of existing α-amylases. Besides truncation and terminal fusion (*Sharma et al., 2021*), these techniques mainly include directed evolution and rational design, which are initially based on in vitro and in silico analyses, respectively. In addition, most enzyme engineering processes will equip both techniques where rational design firstly

identifies a targeted region in the protein sequence based on in silico structural data, which is then mutated randomly via in vitro directed evolution (error prone PCR and DNA shuffling) (*Sharma et al., 2021*).

The tremendous industrial needs have urged the creation of newly engineered α-amylases; however, limited efforts have been paid to uncover the thorough updated information on the published engineering strategies of microbial α-amylases. Here, this review covers mainly the recent advancement on the manipulations of existing microbial α-amylases for enhanced characteristics required by the industries. Therefore, this article is needed to provide better perceptions on the future enzyme manipulation studies with sufficient inputs from the previous works to the researchers in the academia and industries. It also highlights the importance of understanding the three-dimensional structure of α-amylase in enzyme engineering to develop new microbial α-amylases which fit sustainably into the industrial applications.

## Microbial α-amylases: mechanisms of action

α-Amylases catalyze the endo-hydrolysis of starch while β-amylases (E.C. 3.2.1.2; GH 14) catalyze the exo-hydrolysis. This difference in the position of polysaccharide hydrolysis is resulted from the two slightly varied but distinct mechanisms of action, namely retaining (α-amylases; Fig. 1A) and inverting (β-amylases; Fig. 1B), where both involve the displacement(s) of nucleophiles (*Koshland, 1953*; *Zhang, Yip & Withers, 2010*).

α-Amylase has a general five-step reaction mechanism which involves double nucleophilic displacement, giving the popularly known α-retaining double displacement mechanism in hydrolyzing its substrate. There are two key amino acid residues which are usually a pair of acidic amino acids (aspartic acid and/or glutamic acid) separated at an approximately 5 Å distance (*Zechel & Withers, 2000*). The first amino acid acts as a catalytic nucleophile (Fig. 1A) which attacks the anomeric center of the substrate while the second amino acid donates a proton to the glycosidic oxygen at the anomeric center, catalyzing the removal/departure of aglycone (ROH) (*Zhang, Yip & Withers, 2010*; *Van Der Maarel et al., 2002*).

An oxocarbenium ion-like transition state is formed during the glycosylation step before the addition of water molecule and aglycone departure. The proton donation of the second amino acid renders itself as a general base which subsequently catalyzes the second nucleophilic displacement in the deglycosylation step of the covalent glycosyl-enzyme intermediate. Both nucleophilic displacements before glycosylation and after deglycosylation steps involve the oxocarbenium ion-like transition states as aforementioned, thus the α-retaining double displacement mechanism (*Zhang, Yip & Withers, 2010*; *Zechel & Withers, 2001*).

β-Amylases, however, catalyze the hydrolysis of starch into maltose inside the polysaccharide chain (endo-). Although the key catalytic residues are similar with α-amylases, which are commonly a pair of carboxylic acids, these acidic residues are distant from 6–12 Å (*McCarter & Stephen Withers, 1994*; *Zhang, Yip & Withers, 2010*). The more distant acidic amino acids enable the accommodation of the water molecule and substrate at the active site, where the cleavage of scissile glycosidic linkage is performed via general-base

**(A)**

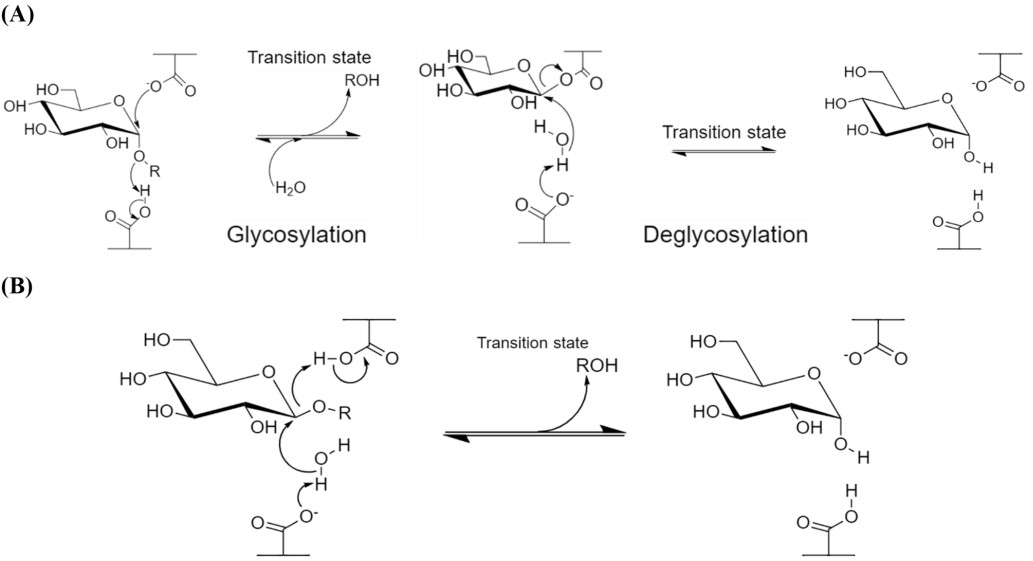

**(B)**

**Figure 1  Reaction mechanisms of amylases.** (A) α-Retaining double displacement mechanism of α-amylases. (B) Inverting single displacement mechanism of β-amylases. The figure was adapted from *Zhang, Yip & Withers, 2010* and all the chemical structures were redrawn using ChemDraw® JS version 19.

catalyzed nucleophilic attack of water molecule on the anomeric center and general-acid-assisted aglycone departure (*Zechel & Withers, 2001*; Fig. 1B). Compared to the retaining mechanism of α-amylases, this inverting mechanism involves only a single nucleophilic step, where the nucleophile from the acidic amino acid attacks the water molecule which subsequently attacks the substrate's anomeric center.

## SURVEY METHODOLOGY

This review was produced based on the research articles published between 2000 and 2020 indexed in the Web of Science Core Collection, Scopus, Elsevier and Google Scholar databases. The search terms used were α-amylase AND (mutagenesis OR engineering) AND (thermostable OR acid OR alkaline OR oxidative OR substrate OR product) AND NOT (nonhuman OR article OR metabolism). The keywords "nonhuman", "article" and "metabolism" were excluded in the literature search because the search results were not relevant to the microbial α-amylases, especially the main scope of this review article. The reference lists of these articles were also screened for relevant papers which could possibly be missed out during the search. Only articles reporting on the engineering of microbial α-amylases were included and discussed. Three exceptional articles published before the year 2000 were also included for the important research outputs. This review provides an overview on the advances of modifying microbial α-amylases via protein engineering techniques to achieve longer half-life in high temperature, improved resistance (acidic, alkaline and oxidative) as well as enhanced substrate and product specificities.
## Microbial α-amylases for industrial applications

Most industrial important enzymes especially α-amylases are preferably isolated from microbial sources encompassing mainly the bacteria, yeasts and fungi due to the ease in genetic manipulations and bulk productions (*Lim, Oslan & Oslan, 2020*). Therefore, native and recombinant α-amylases have frequently been produced by or cloned into various microbial expression hosts including *Bacillus subtilis* (*Trabelsi et al., 2019a*), *Enterococcus faecalis* (*Meruvu, 2019*), *Aspergillus clavatus* (*Shruthi, Achur & Boramuthi, 2020*), *Tepidimonas fonticaldi* (*Allala et al., 2019*), *Komatagaella phaffii* (*Trabelsi et al., 2019b*; *Wang et al., 2019*) and *Meyerozyma guilliermondii* (*Nasir et al., 2020*).

## Microbial enzymes production and purification

α-Amylases can be produced extracellularly or intracellularly by the microorganisms. While most heterologous proteins produced intracellularly in the obsolete *Escherichia coli*-based expression system form the insoluble fractions called inclusion bodies (IBs) (*Singhvi et al., 2020*), the addition of signal peptides in the recombinant plasmids can resolve such issue in most bacteria. However, yeasts are favoured for industrial production of α-amylases due to their simplicity as unicellular organisms with more similar post-translational modifications (PTMs) with higher level eukaryotes, where an in vitro process is needed for PTMs in bacterial expression systems (*Vieira Gomes et al., 2018*).

From the microbial fermentation products, the cells are usually separated from their spent media via centrifugation at cold temperature (4 °C). The extracellular α-amylases which are secreted into the fermentation media will present in the supernatants. The pelleted cells need to be ruptured to release the intracellular α-amylases where the most frequently used method is sonication compared to others (ball milling, enzymatic cell wall removal, freezing-thawing, liquid shearing and osmotic shock) (*Robinson, 2015*). The ruptured cells are again pelleted, and the intracellular α-amylases are extracted in the supernatants.

Following enzymes extraction, (partial) purifications are performed to yield the pure α-amylases which are readily applied in industries. A recent study has reported on the usages of ammonium sulfate precipitation and anion exchange chromatography (AEX) using diethyl amino ethyl (DEAE) cellulose resin to purify a thermostable α-amylase expressed in *Bacillus* sp. strain SP-CH7 with a 65.95-fold purification and 25.9% yield (*Priyadarshini & Ray, 2019*). Ammonium sulfate, being the most frequently equipped salt to precipitate enzymes, is added slowly into the crude enzymes to increase the hydrophobic interaction between α-amylases and water with decreasing contact surface area of the enzymes with the solvent (water) (*Lim, Oslan & Oslan, 2020*).

However, desalting step via dialysis is required to remove excessive salt from the partially purified α-amylases for more accurate characterizations. Interestingly, Feng's research group (*Xian et al., 2015*) performed desalting via HiPrep 26/10 desalting column, resulting in higher enzyme recovery (80.13%) compared to the dialysis-based desalting technique. While plant α-amylases tend to be in two isoforms i.e., high and low isoelectric point (pI) (*Ju et al., 2019*), most microbial α-amylases only exhibit low pI isoforms since the resins reported for microbial α-amylases purification are anionic exchangers (DEAE

and Q) (*Sudan et al., 2018*; *Priyadarshini et al., 2020*). α-Amylases can also be purified via ultrafiltration and gel filtration chromatography albeit salt precipitation and AEX.

Gel filtration chromatography (also known as size-exclusion chromatography) is usually equipped after salt precipitation and AEX as a desalting step despite its primary function to purify α-amylases based on their sizes. Two-time ultrafiltration (using 100,000 and 30,000 molecular weight cut off membranes) and Sephadex G-100 were performed to purify a polyextremotolerant α-amylase expressed in *E. faecalis* mercadA7 with 24% yield at 7.2-fold purification (*Meruvu, 2019*). Interestingly, a recent study has reported on the purification of three native α-amylases (known as Amy586) in *B. subtilis* strain US586 through a three-step purification encompassing heat treatment (30 min at 55 °C), acetone concentration and gel filtration (Superdex 200) with the final yield of only 10% with 92.7-fold purification (*Trabelsi et al., 2019a*).

Moreover, affinity chromatography, perhaps, is the most preferred purification method when α-amylases are expressed recombinantly. Despite the large size of glutathione S-transferase (GST) tag i.e., 26 kDa which is commonly used in *E. coli*-based expression vector (pGEX) (*Kimple, Brill & Pasker, 2013*), polyhistidine (His$_6$) tag is preferably employed in immobilized metal affinity chromatography (IMAC) of α-amylases. In 2015, Gandhi et al. reported on the single-step purification of His-tagged SR74 α-amylase expressed in *K. phaffii* GS115 (previously known as *Pichia pastoris*) using five mL HiTrap IMAC FF with 52.6% yield at 1.9-fold purification (*Gandhi et al., 2015*).

Surprisingly, hydrophobic interaction chromatography (HIC) has also been performed to purify fungal and bacterial α-amylases using HiPrep 16/10 phenyl FF (*Xian et al., 2015*) and Phenyl-Sepharose column (*Keskin & Ertunga, 2017*), respectively. Although a variety of methods are available to purify microbial α-amylases, the ultimate aims of purifications are to achieve both high recovery yield and purification fold, besides providing the most accurate characterization of α-amylases for industrial applications.

## Enzymes characterizations for industrial applications

Several microbial α-amylase characterizations have been reported, encompassing optimum temperature, optimum pH, thermal stability, pH stability and tolerances (salinity, detergents, and organic solvents). These characterizations are essential to evaluate and determine the applicability of the concerned α-amylases in the wide industrial fields. In food and beverages industries, α-amylases have been used in starch processing (saccharification and liquefaction; starch-to-ethanol conversion), break making, high fructose corn syrup (HFCS) formation, beer brewing and haze clarification in fruit juice.

α-Amylases which involve in starch saccharification and liquefaction must be thermostable or hyperthermostable due to the elevated temperature during these processes (*Lim, Oslan & Oslan, 2020*). The hydrolysis rates of 1% (w/v) corn, wheat and potato starches at 16 h (60 °C, pH 7) were 57.52, 49.61 and 32.35% respectively when 1 U/mg starch of purified α-amylase expressed in *B. amyloliquefaciens* BH072 was used (*Du et al., 2018*; Table 1). Interestingly, a more powerful thermostable α-amylase from *Geobacillus* sp. K1C recorded the degradation rates of 10% (w/v) rice, wheat and potato starches at
88.1%, 90.3% and 81.1% respectively after 4 h incubation with 0.1 U/mg enzyme at 80 °C and pH 6 (*Sudan et al., 2018*).

Besides, three acid-stable α-amylases isoforms expressed in *B. subtilis* strain US586 has been assessed for its application in the bread making industry (*Trabelsi et al., 2019a*). The α-amylases (Amy586) supplementation (0.06 U/g) in flour not only increased the dough quality (elasticity: extensibility ratio = 0.5 –1.2), but also significantly decreased the bread hardness with enhanced cohesion and elasticity (*Trabelsi et al., 2019a*). The intermediate temperature stable (ITS) *Laceyella* sp. DS3 α-amylases (AmyLa; wild type and recombinant) were said to be desirable in the baking industries as an antistaling agent while exhibiting the optimum temperature at 50 °C and 55 °C, respectively (*El-Sayed et al., 2019*). Nevertheless, α-amylase supplementation was also expected to improve the texture, flavour and shelf-life of the bread. In addition, a *Thermomyces dupontii* α-amylase (TdAmyA) expressed in *K. phaffii* was promising in maltose syrup production since the highest maltose percentage (approximately 52%) could be achieved in a short hydrolysis time of liquefied starch at 8 h (*Wang et al., 2019*).

In the beverage industry, α-amylases have been used to clarify fruit juices since a dark product may not be preferred by the consumers (*Uzun, Demirci & Akatin, 2018*). A study had reported on the reduction of colour intensity of the raw apple juice ($OD_{440nm}$) from 1.537 to 0.443 after 3 h incubation of 4.9 mL pasteurized apple juice with 0.1 mL (1 mg/mL) purified α-amylase expressed in *Rhizoctonia solani* AG-4 strain ZB-34 (*Uzun, Demirci & Akatin, 2018*). Captivatingly, a metatranscriptomics study of Chinese Nong-flavour liquor starter has successfully mined a fungal extracellular thermostable α-amylase (NFAmy13A) which exhibits optimum activity at 60 °C and pH 5.0–5.5 (*Yi et al., 2018*). In addition, its prominent role in beer brewing and liquor starter had been evident with its highest substrate specificity towards amylopectin which was more abundantly detected in wheat material compared to amylose (*Yi et al., 2018*).

In detergent industry, the desirable traits of α-amylases are more complex than other industries, encompassing being thermostable, alkaline-stable and stable towards various detergent components (oxidizing agents and surfactants). The crude α-amylase expressed in *B. mojavensis* SA was proven to be tolerant towards sodium dodecyl sulfate (SDS), Tween 20, Tween 80, Triton X-100, sodium perborate as well as reactive hydrogen peroxide ($H_2O_2$) (*Hammami et al., 2018*). Nevertheless, the starch stain removal rate of purified alkaline α-amylase (AA7) expressed in *Bacillus* sp. strain SP-CH7 reduced insignificantly from 96.083% to 95.960% when detergent was added, proving its compatibility in detergent industry while maintaining the brightness and softness of the tested cotton fabrics (*Priyadarshini & Ray, 2019*). Notably, a cold-adapted, halophilic α-amylase (Amy175) from Antarctic sea ice bacterium *Pseudoalteromonas* sp. M175 has been evident for its primary application in detergent industry where it retained more than 76.9% of amylolytic activity after one hour of incubation at 25 °C with all commercial laundry detergents tested (*Wang et al., 2018a*). It is also reported that the addition of Amy175 (12.6 U/mL) with detergent exhibited better wash performance than the sole detergent, proving its huge potential as the detergent additive (*Wang et al., 2018a*).

**Table 1  Characterization of microbial α-amylases and their desired traits for industrial applications.**

| Sources of α-amylases | Optimum temperature and pH | Desired traits and performance | References |
|---|---|---|---|
| Starch Saccharification and Liquefaction | | | |
| *Bacillus amyloliquefaciens* BH072 | 60 °C, pH 7.0 | Hydrolyzed 57.52%, 49.61%, and 32.35% of 1% (w/v) corn, wheat and potato starches, respectively | *Du et al. (2018)* |
| *Geobacillus* sp. K1C | 80 °C, pH 6.0 | Degraded 88.1%, 90.3%, and 81.1% of 10% (w/v) rice, wheat, and potato starches, respectively | *Sudan et al. (2018)* |
| Baking | | | |
| *Bacillus subtilis* strain US586 | 60 °C, pH 4.0–6.0 | Decreased elasticity: extensibility ratio to 1.2; increased dough baking strength to $172 \times 10^{-4}$ J | *Trabelsi et al. (2019a)* |
| *Laceyella* sp. DS3 (expressed in *Escherichia coli* BL21) | 55 °C (intermediate temperature stable, ITS), pH 6.0 -7.0 | Not stated | *El-Sayed et al. (2019)* |
| Beverages | | | |
| *Thermomyces dupontii* (expressed in *Komagataella phaffii*) | 60 °C, pH 6.5 | Converted 52% liquefied starch to maltose at 8 h | *Wang et al. (2019)* |
| *Rhizactonia solani* AG-4 strain ZB-34 | 50 °C, pH 5.5 | Reduced colour intensity of raw apple juice ($OD_{440nm}$) up to 71.2% at 3 h | *Uzun, Demirci & Akatin (2018)* |
| Fungus (expressed in *K. phaffii*) | 60 °C, pH 5.0–5.5 | Maximum specific activity (200.4 U/mg) on amylopectin which is abundantly found in wheat material | *Yi et al. (2018)* |
| Detergent | | | |
| *Bacillus mojavensis* SA | 55 °C, pH 9.0 | Retained >34% activity in 1% non-ionic and anionic surfactants as well as oxidizing agents | *Hammami et al. (2018)* |
| *Bacillus* sp. SP-CH7 | 65 °C, pH 9.0 | Retained >95% starch stain removal rate with detergent added | *Priyadarshini & Ray (2019)* |
| *Pseudoalteromonas* sp. M175 | 25 °C, pH 8.0 | Retained >76.9% amylolytic activity towards tested laundry detergents with better wash performance as a detergent additive | *Wang et al. (2018a)* |
| Textile and Leather | | | |
| *Aspergillus luchuensis* BS1 | 60 °C, pH 5.5 | Desized cotton fabric with 9.5% weight loss, 5 s of absorbency time and 8 rating in Tegewa analysis | *Sadhasivam et al. (2018)* |
| Biodegradation | | | |
| *B. subtilis* TB1 | Not stated | Increased biodegradation efficiency (53%) of residual hydrocarbons in the presence of starch | *Karimi & Biria (2016)* |
| | | Reduced weight (48%) and tensile strength (87%) of low-density polyethylene (LDPE)-starch blend samples | *Karimi & Biria (2019)* |

While coating (sizing) with a gelatinous substrate (size) like starch is required to prevent breakage during the weaving process in the textile industry, desizing (removal of size-like starch) is needed to allow water and finishing agents absorbances without fiber damage (*Mehta & Satyanarayana, 2016*). A cotton fabric desized with 1.8% partially purified α-amylase expressed in *Aspergillus luchuensis* BS1 was found to exhibit 9.5% weight loss when treated at 60 °C for 1 h as well as 5 s of wetting time and 8 rating in Tegewa analysis while employing iodine drop test (*Sadhasivam et al., 2018*).

Nonetheless, the α-amylase expressed in *B. subtilis* TB1 had been reported to involve in the biodegradation and bioremoval of n-alkanes (in 1% v/v n-paraffins) with 53% reduction in residual hydrocarbons in the system with the presence of starch, a carbohydrate polymer (*Karimi & Biria, 2016*). In addition, the subsequent study (*Karimi & Biria, 2019*) has evident the commercial α-amylase was able to reduce the weight (48%), tensile strength (87%) of low-density polyethylene (LDPE)-starch blend samples as well as molecular weight (70%) and viscosity (60%) of LDPE, thus exhibiting a promiscuous cometabolic effect while biodegrading LDPE in the blends. Besides biodegradation, α-amylases have also been used in feed industry to improve digestibility of animal feeds.

Importantly, the engineering of α-amylase to improve its suitability in industrial applications is highly guided with its structural and sequence analyses despite the sole functions of directed evolution and rational design. Therefore, the understanding on the structural properties of α-amylases is of utmost important before performing enzyme engineering for its biotechnological traits and benefits.

## Structural properties of microbial α-amylases

The structure of a protein is more conserved compared to its sequence after evolutions. Hence, the dissection on microbial α-amylase structural properties is crucial and essential prior to designing an existing enzyme with enhanced characteristics. As aforementioned, α-amylases have been grouped into 4 distinct families encompassing GH 13, GH 14, GH 57 and GH 119 with GH 13 accommodating highest number of α-amylases. Therefore, this section will discuss only on the structural properties of microbial α-amylases which fall in GH13 in CaZy database based on a recently elucidated structure of *B. paralicheniformis* ATCC 9945a α-amylase (*Bli* Amy) (*Božić et al., 2020*).

A typical α-amylase has a classical three-domain fold, namely Domain A (residues 3–103 and 206–396), Domain B (residues 104–205) and Domain C (residues 397–482) (Fig. 2A). N-terminal domain A, where the catalytic triad residues (D231, E261 and D328) were located, was also known as the catalytic domain possessing a central (β/ α)$_8$ TIM-barrel structure. Notably, domain B was extruded from domain A, consisting of two extended loops. The C-terminal domain C was, however, composed of a β-barrel with 8 antiparallel strands. This domain consisted of a Greek key motif which was attributed to substrate binding in *Geobacillus thermoleovorans* α-amylase (*Mehta & Satyanarayana, 1994*).

Furthermore, several surface-binding sites (SBS) were present in *Bli* Amy despite having carbohydrate-binding module (CBM) as another starch-binding domain (SBD) reported. However, only the SBS involving maltotetraose (G4) binding is selected and shown in Fig.

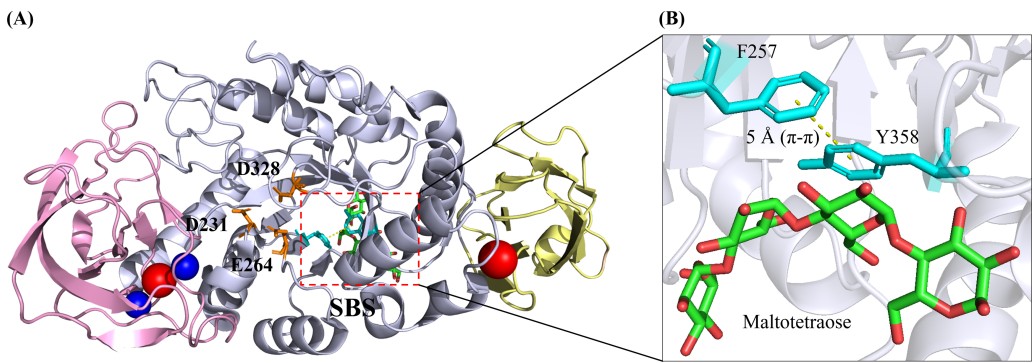

**Figure 2** **3D crystal structure of *B. paralicheniformis* α-amylase (*Bli* Amy; PDB ID: 6TP1) with maltotetraose at its substrate-binding site (SBS) (*Božić et al., 2020*).** (A) *Bli* Amy has a three-domain fold structure which is composed of Domain A, Domain B and Domain C as depicted in light blue, violet and wheat colours, respectively. The catalytic triad residues (D231, E264 and D328) are shown as orange sticks while the calcium and sodium ions are depicted as blue and red balls, respectively. (B) The SBS in *Bli* Amy consists of two hydrophobic residues (F257 and Y358) in cyan colour where the yellow dashed line between the residues is the π- π hydrophobic interaction at 5 Å. Maltotetraose (MTT) molecule is shown as green carbon backbone. The structural image was generated using the PyMOL Molecular Graphic System, Version 2.4, Schrödinger, LLC.

2A. Unlike CBM located usually at separate domains which connected to the N- or C-termini of the catalytic domain (Domain A) via a polypeptide linker, SBS was a non-catalytic site present mostly in the catalytic domain, which enhanced the starch-absorptivity level and subsequently increased the raw starch degradation rates (*Baroroh et al., 2017*; *Božić et al., 2020*). In *Bli* Amy, the SBS which binds maltotetraose has two key amino acids F257 and Y358 (Fig. 2B). They provide a hydrophobic platform for the carbohydrates while their substitutions with alanine resulted in 5-fold lower raw starch catalytic efficiency with more than 5.5-fold lower affinity compared to the wild type (*Božić et al., 2020*).

The structural function of SBS has been validated when α-amylase from *Saccharomyces fibuligera* R64 (Sfamy R64) was shown to exhibit low starch adsorptivity due to the absence of SBS (*Yusuf et al., 2017*). In addition, an acarbose molecule which was a typical inhibitor of α-amylases, was also observed to bind in the SBS of *B. stearothermophilus* STB04 (Bst-MFAse; PDB ID: 6ag0) located in the active site (*Xie et al., 2019a*). Importantly, a pair of aromatic amino acid residues was found at each of the three SBSs in the *Halothermothrix orenii* α-amylase (AmyB), where the hydrophobic CH/π-stacking interaction was prominently observed (*Tan et al., 2008*). Tetrasaccharide, β-cyclodextrin, and glucose were bound to SBS I (W488 and Y460), SBS II (W287 and W260) as well as SBS III (W310 and W306), respectively (*Tan et al., 2008*).

To date (March 2021), there have been 88 different CBM families established in the CAZy database (http://www.cazy.org/Carbohydrate-Binding-Modules.html) whereby 12 CBM families have been reported in the α-amylases (E.C. 3.2.1.1) (*Lombard et al., 2014*). CBMs have been recognized as SBD in starch-hydrolyzing enzymes which include microbial α-amylases (*Baroroh et al., 2017*), and most CBMs adopt a β-sandwich fold which is carbohydrates-binding (*Janecek et al., 2019*). The removal of CBMs in *Eubacterium rectale*
(a butyrate-producing gut bacterium) α-amylase has ≈40-fold reduction in its enzymatic activity towards corn starch, highlighting the presence of CBMs in improving the catalytic function of the α-amylase (*Cockburn et al., 2018*).

Interestingly, there are three types of CBMs classified based on their topology of the ligand-binding site, namely Type A (planar and hydrophobic surface which binds insoluble carbohydrates encompassing crystalline cellulose and chitin), Type B (variable loop sites, VLS and concave face sites, CFS which bind various glycans such as starch, mannans, galactans and xylans) as well as Type C (pocket which recognizes short-chain or exposed carbohydrates especially 1C-, 2C-, and 3C-carbohydrates) (*Armenta et al., 2017*; *Campos et al., 2016*). These interactions between the binding sites (CBM and/or SBS) are generally made up of hydrophobic interactions (CH-π) which involve the strictly conserved tryptophan, tyrosine and phenylalanine residues (*Vujičić-Žagar & Dijkstra, 2006*), and subsequently supported by the formation of hydrogen bonds, especially in CBM Type B and C (*Campos et al., 2016*). However, the presence and number of CBMs in microbial α-amylases vary within organisms despite the classical three-domain fold (*Sidar et al., 2020*).

While certain loops and secondary structures might contribute to the physicochemical properties of the enzyme, it is noteworthy that the protein-ligand interactions also played indispensable roles in stabilizing the α-amylase structure. As illustrated in Fig. 2A, *Bli* Amy possessed four metal ion ligands: two calcium ions ($Ca^{2+}$) and two sodium ions ($Na^{+}$) (*Božić et al., 2020*). Two calcium ions (CaI-CaII) and a sodium ion (NaI) formed a metal triad which was in the interior of Domain B, while NaII resided between Domain A and Domain C.

Despite the often-reported $Ca^{2+}$-$Na^{+}$-$Ca^{2+}$ linear metal triad (*Offen et al., 2015*), Bst-MFAse had the unprecedented $Ca^{2+}$- $Ca^{2+}$- $Ca^{2+}$ triad (*Xie et al., 2019a*), where it had been used as the template to predict the 3D structures of both wild type and recombinant bacterial SR74 α-amylases through homology modeling strategy (*Lim et al., 2020*). However, NaII in *Bli* Amy replaced the calcium ion which had been reported to be located between Domain A and Domain C, where its role was not described (*Božić et al., 2020*). On the contrary, the calcium ion (CaIV) in SR74 α-amylase was hypothesized to involve in substrate specificity and catalytic activity of the enzyme, by bridging Domain A and Domain C (*Lim et al., 2020*).

Besides the wide availability of α-amylases sequences, the understanding of their structural properties should not be underexplored. In facts, protein engineering relies heavily on both sequential and structural information of the existing α-amylases, by means of directed evolution, rational design or the combination of both (*Sharma et al., 2021*).

## Engineering of microbial α-amylases for industrial applications

The processes in the industries often involve harsh conditions. Several important features encompassing thermostability, pH tolerance, substrate and product specificities as well as oxidative stability, are therefore, favoured in the industries. Being the industrial important enzymes, α-amylases have extensively been studied in terms of their categorizations, sequences and structures. The ample knowledge on α-amylases have allowed researchers to

modify and engineer various native and recombinant α-amylases for their beneficial traits in the industrial applications (Table 2).

## Engineered microbial α-amylases with increased thermostability

Over the decades, various microbial α-amylases have been engineered for elevated thermostability via several methods including directed evolution, rational design and others. These studies which have evident the indispensable roles of disulphide linkages, hydrogen bonds, metal ions, salt bridges, non-polar interactions and stabilization or removal of flexible or extended loops in α-amylases resistance towards thermal inactivation will be described and discussed.

A site-directed mutagenesis (SDM) in a novel *Flavobacteriaceae sinomicrobium* α-amylase (FSA) had been performed to introduce a disulphide bridge in its Domain C based on the multiple sequence alignment (MSA) with α-amylases in another clade of *Flavobacteriaceae* genus (*Li et al , 2014*). By substituting both K415 and S450 in the wild type FSA with cysteines, elevated half-life ($t_{1/2}$) at 50 °C from 25 to 55 min with an optimum activity at 55 °C was observed besides the thermal inactivation reversal at 100 °C with more than 50% residual activity detected for the mutant FSA (*Li et al , 2014*). The stability retainment under the extreme conditions of mutant FSA was preferred in the industrial production of food, paper, bioethanol, textiles and detergents although the promising results on these applications were not established in this study.

Besides the intra-domain disulphide bridge, an inter-domain disulphide linkage was introduced between the Domain A and Domain C of a yeast *Saccharomycopsis fibuligera* R64 α-amylase expressed in *K. phaffii* KM71H (Sfamy01) by replacing both S336 and S437 with cysteine residues (*Natalia et al., 2015*). Albeit the suggested mutation candidates were S336 and S478, the location of S478 in a β-turn and its stabilizing interactions with the β-strands had led to the mutation of another loop-residing serine (S437) which did not contribute to β-sheets arrangement. The mutant α-amylase (Sfamy02) with improved hydrophobic interaction between two domains was shown to maintain 60% of its residual activity compared to the substantial decline in Sfamy01, therefore suitable for raw-starch degradation in the rice and cassava fermentation (*Natalia et al., 2015*).

In spite of the increased activity (3.2%) detected by introducing a new disulphide linkage in FSA at E200C and H201C, enhanced ligand-dependent thermostability i.e., $Ca^{2+}$ and $Zn^{2+}$ was detected after the incubation at 50 °C for 30 min when E204G and C214S mutations were performed (*Yin et al., 2017*). Nonetheless, the improvement in thermostability ($t_{1/2}$ at 95 °C) at 26-fold ($Ca^{2+}$-absent) and 5-fold ($Ca^{2+}$-present) was observed when the loop deletion of I181 and G182 (ΔIG) coupled with N193F and S242A were performed on the maltohexaose-forming α-amylase from *B. stearothermophilus* (AmyMH) (*Li, Duan & Wu, 2016*).

These findings further evident that the enhancement of ligand binding and flexible extended loop deletion may improve thermostability of the α-amylases while suggesting the mutant AmyMH (ΔIG/N193F/S242A) is promising for industrial starch liquefaction. In addition, loop deletion (ΔR179-G180) was also proven to enhance the thermostability

**Table 2** Engineering of microbial α-amylases for enhanced thermostability, pH tolerance, substrate and product specificities as well as oxidative stability.

| Microbial strains | Modifications | Improvements | References |
|---|---|---|---|
| Thermostability | | | |
| B. subtilis CN7 | V260I | Increased melting point (7.1 °C) and half-inactivation temperatures (4.9 °C) | Wang et al. (2020) |
| B. stearothermophilus | ΔR179 - G180 | Increased half-life at 100 °C (24 to 33 min) | Gai et al. (2018) |
| Flavobacteriaceae sinomicrobium | E200C, H201C | Increased ligand-dependent thermostability at 50 °C for 30 min | Yin et al. (2017) |
| B. licheniformis | S187D, N188T, A269K | Increased half-life (9-fold) at 95 °C | Li et al. (2017) |
| B. stearothermophilus | ΔI181 –G182, N193F | Increased half-life at 95 °C at 26-fold ($Ca^{2+}$-absent) and 5-fold ($Ca^{2+}$-present) | Li, Duan & Wu (2016) |
| B. acidicola (Ba), G. thermoleovorans (Gt) | Chimeric Ba-Gt | Increased half-life at 90 °C (5 to 15 min) | Parashar & Satyanarayana (2016) |
| Saccharomyces fibuligera R64 | S336C, S437C | Maintained 60% activity at 65 °C | Natalia et al. (2015) |
| Geobacillus sp. SK70 | Q294H | Increased optimum temperature (55 to 60 °C) and thermostability (35 to 85 min) | Sulong et al. (2015) |
| F. sinomicrobium | K415C, S450C | Increased half-life at 50 °C (25 to 55 min); maintained >50% activity at 100 °C | Li et al (2014) |
| E. focardii | E166P, S185P, V212T, V232T, T350P | Increased optimum temperature (25 to 30 °C) and thermostability at 50 °C (1.8 to 3.3 min) | Yang et al. (2017) |
| pH tolerance and stability | | | |
| Rhizopus oryzae | V174R | Increased half-life (2.55-fold) at pH 4.5 | Li, Yang & Tang (2020) |
| B. licheniformis | G81R | Maintained 10% activity at pH 4.5 for 40 min | Huang et al. (2019) |
| B. subtilis | A270K, N271H | Decreased in optimum pH (pH 6.5 to 4.5); maintained >75% activity at pH 3.5 | Wang et al. (2018a); Wang et al. (2018b) |
| B. stearothermophilus | ΔR179 –G180 | Increased acid-resistance in range of pH 4.5 –6.0; decreased optimum pH (pH 5.5) | Gai et al. (2018) |
| Rhizopus oryzae | H286E | Increased half-life (6.43-fold) at pH 4.5 (57.28 to 66.65 min); decrease optimum pH (pH 4.5) | Li et al. (2018) |
| B. licheniformis | H293R, H316R, H327R | Maintained 31% activity at pH 4.5 for 40 min | Liu et al. (2017) |
| Alkalimonas amylolytica | H209L, Q226V, P477V | Increased optimum pH (pH 10.0) with active pH range (pH 6.0 –12.0) | Deng et al. (2014) |

**Table 2** (*continued*)

| Microbial strains | Modifications | Improvements | References |
|---|---|---|---|
| Substrate and product specificities | | | |
| *Talaromyces leycettanus* JCM12802 | CBM20-linker substitution (7 to 21 residues) | Increased (>65.3%) substrate specificities in vitro | *Zhang et al. (2017)* |
| | Y401W | Increased ($\approx$ 10%) substrate specificities in vitro | *Amalia et al. (2016)* |
| *Saccharomycopsis fibuligera* R64 | S383Y, S386W | Improved (86.5%) substrate affinity at 20 ns in silico | *Yusuf et al. (2017)* |
| | S383Y, S386W, N421G, S278N, A284K, Q384K, K398R, G400_S401insTDGS | Improved (29.3%) substrate affinity at 100 ns in silico | *Baroroh et al. (2019)* |
| *B. subtilis* CN7 | Y204(F, I), V260(I, L) | Increased glucose (G1) production than maltose (G2) | *Wang et al. (2020)* |
| | G109(N, D, F) | Increased maltohexaose (G6) production (36.1, 42.4, 39.0% respectively) | *Xie et al. (2020)* |
| *B. stearothermophilus* | W139(A, L, Y) | Increased maltopentaose (G5) production | *Xie et al. (2019a)*; *Xie et al. (2019b)* |
| *Rhizopus oryzae* | H286(L, M) | Increased affinities towards maltotriose (G3) and soluble starch; increased G2 production | *Li et al. (2018)* |
| *B. stearothermophilus* | W177(F, Y, L, N, S) | Reduced G3 formation through increased hydrolysis of soluble starch | *Sun et al. (2016)* |
| Oxidative stability | | | |
| | M145I-214A-229T-247T-317I | Improved (5.4-fold) and maintained 91.3% activity in 500 mM $H_2O_2$ for 1 h | *Yang et al. (2013a)* |
| *Alkalimonas amylolytica* | Fusion of oligopeptide at N-terminal | Improved (2.7-fold) and maintained 54% activity in 500 mM $H_2O_2$ for 30 min | *Yang et al. (2013b)* |
| *Thermotoga maritima* | M55A, (M43A, M44A) | Maintained 50% and 39% activity in 100 mM $H_2O_2$, respectively | *Ozturk et al. (2013)* |
| *Alkalimonas amylolytica* | M247L | Improved oxidative resistance (72%) | *Yang et al. (2012)* |
| *Bacillus* sp. strain TS-23 | Truncated C-terminal, M231L | Maintained >96% activity in 500 mM $H_2O_2$ | *Chi et al. (2010)* |
| *G. stearothermophilus* US100 | $\Delta$I214 –G215, M197A | Maintained 70% activity in 1.8 M $H_2O_2$ for 60 min | *Khemakhem et al. (2009)* |

of another α-amylase from *B. stearothermophilus* (AmyS) by increasing the $t_{1/2}$ at 100 °C from 24 min to 33 min (*Gai et al., 2018*).

Nevertheless, a 9-fold improvement on the thermostability ($t_{1/2}$ at 95 °C) of *B. licheniformis* α-amylase was recorded by increasing number of hydrogen bonds and salt bridges through the mutations of loop residues (S187D and N188T) and an adjacent residue (A269K) (*Li et al., 2017*). Such strategy was also equipped by a previous study when the Q294H mutation within the catalytic domain (Domain B) of *Geobacillus* sp. SK70 α-amylase was performed to increase the optimum temperature (55 °C to 60 °C)

and thermostability ($t_{1/2}$ at 60 °C from 35 min to 85 min) based on a semi rational resign approach (*Sulong et al., 2015*).

In addition, a rationally engineering strategy was performed to introduce V260I mutation near to the central β-strands of *B. subtilis* CN7 α-amylase's TIM barrel (*Wang et al., 2020*). The increased melting ($T_m$; 7.1 °C) and half-inactivation temperatures (4.9 °C) were attributed to the extra 8 weak *Van de Waals* (VDW) contacts within the central β-strands (*Wang et al., 2020*). The choices of both mutation candidate and replacing residue are crucial to introduce certain polar or non-polar linkages through mutation, where the comprehensive knowledge on amino acid residues is essential.

Notably, a cold-adapted but thermolabile α-amylase expressed by the Antarctic ciliate protozoon *Euplotes focardii* (*Ef* Amy) had been mutated at the surface loops of Domain A (T350P) and B (E166P/ S185P) as well as the residues (V212T/V232T) near to the catalytic site (*Yang et al., 2017*). Such SDM was multiple sequence alignment (MSA)-assisted where both mesophilic *Euplotes crassus* α-amylase (*Ec* Amy) and *A. oryzae* TAKA α-amylase were compared for protein secondary structure analysis. The mutant with combined SDM exhibited higher optimum temperature (30 °C) compared to its wild type (25 °C) with 1.8-fold increase in its $t_{1/2}$ at 50 °C, without affecting its optimum pH (*Yang et al., 2017*). These substitutions to proline and valine residues had conferred a higher rigidity to the thermolabile *Ef* Amy through the introduction of more polar hydrogen bonds. Such enhanced properties of rational-designed *Ef* Amy might favour its usage in detergent, food and textile industries, which worth to be explored and evident further by the group of researchers.

Interestingly, chimerism which involves truncation and terminal fusion is perhaps the most special engineering strategy compared to the common directed evolution and rational design. A chimeric α-amylase created using the truncated *B. acidicola* (Ba-amy) with the partial N- and C-termini of *G. thermoleovorans* (Gt-amy) α-amylases was proven to exhibit higher thermostability and catalytic efficiency (*Parashar & Satyanarayana, 2016*). The acquisition of a better functioning starch binding domain (SBD) from the C-terminal of Gt-amy and the increase in β-sheet content in the circular dichroism (CD) results of the third chimer (ch3) might contribute to the improved catalytic efficiency and the increase of $t_{1/2}$ from 5 min to 15 min at 90 °C when expressed in *E. coli* (*Parashar & Satyanarayana, 2016*). Captivatingly, a 10.7-fold increment in α-amylase (Ba-Gt-amy) titre expressed in a multi-copy *K. phaffii* clone and its attractive physicochemical properties have made it suitable for baking and sugar syrup industries (*Parashar & Satyanarayana, 2017*).

## Engineered microbial α-amylase with pH tolerance and stability

Although rational design has frequently been used to engineer the microbial α-amylases for better pH tolerance and stability, error prone PCR (epPCR) and site-directed mutagenesis (SDM) are also performed to achieve the similar effect. An epPCR was conducted by Liu's group (*Huang et al., 2019*) to generate a mutant library with 2,300 clones, where G81R was found to retain 10% of its residual activity after incubating at pH 4.5 for 40 min when the wild type was already inactivated. The mutation from glycine (non-polar) to arginine

(positively charged) in *B. licheniformis* α-amylase (BLA) stabilized the negatively charged D231 and decreased its pKa (*Huang et al., 2019*).

The pKa reduction in the nucleophilic catalytic residue (D174) was also observed when N271H was introduced to *B. subtilis* Ca-independent α-amylase (Amy7C) (*Wang et al., 2018b*). This study by Zhao's group (*Wang et al., 2018b*) had utilized both rational design and site-directed mutagenesis and subsequently evident that A270K/N271H double mutant exhibited 2-unit decline in optimum pH (pH 4.5) with approximately 3.94-fold improvement in catalytic efficiency. In addition, the pH range at the acidic limb was found to be shifted to pH 3.5 with more than 75% residual activity.

Similar strategy to reduce the pKa of nucleophile and hydrogen donor in catalytic residues was performed through the SDM of H293R/H316R/H327R in BLA, resulting in the triple mutant's ability to maintain 31% of residual activity at pH 4.5 and 70 °C for 40 min (*Liu et al., 2017*). Interestingly, loop deletion is not only capable of increasing the enzyme thermostability, but a study concerning the deletion of the flexible loop (ΔR179-G180) in *B. stearothermophilus* α-amylase (AmyS) also showed its ability to improve the acid-resistance of AmyS to pH 4.5−6.0 with decreased optimal pH at pH 5.5 (*Gai et al., 2018*).

Nevertheless, the formation of addition hydrogen and salt linkages might also improve the acid resistance of an α-amylase besides its thermostability. This was proven when a MSA-based V174R SDM was introduced to a yeast *Rhizopus oryzae* α-amylase (ROAmy), whereby the mutant exhibited an improvement in thermostability ($t_{1/2}$ at 55 °C), pH resistance ($t_{1/2}$ at pH 4.5) and catalytic efficiency ($k_{cat}/K_m$) on soluble starch at 2.52-, 2.55- and 1.61-fold, respectively (*Li, Yang & Tang, 2020*). In addition, a previous site-saturation mutagenesis was also performed on H286 of ROAmy (*Li et al., 2018*). The mutant H286E exhibited a 6.43-fold improvement in half-life (from 57.28 min to 66.65 min) at pH 4.5, which was correlated to its reduced optimum pH from pH 5.5 (wild type) to pH 4.5 (H286E mutant) (*Li et al., 2018*).

These engineered microbial acid-resistant α-amylases are promising and suitable for starch liquefaction and saccharification industries since both processes occur at different pH ranges (pH 5.8−6.2 and pH 4.2−4.5, respectively) (*Wang et al., 2018b*). Albeit the importance of alkaline-resistant α-amylases in detergent industry, it is noteworthy that the latest microbial α-amylase engineering for its alkaline tolerance was reported by Chen's group (*Deng et al., 2014*). The group applied structure-based rational design for systems engineering of *Alkalimonas amylolytica* α-amylase, whereby the triple mutant (H209L/Q226V/P477V) exhibited higher optimum pH (pH 10.0) and broader pH range (pH 6.0–12.0) compared to its wild type (pH 9.5; pH 7.0–11.0) (*Deng et al., 2014*).

Nonetheless, these studies have successfully proven that the usage of rational design, directed evolution or combination of both are powerful to modify, engineer and generate new α-amylase mutants with enhanced acid or alkaline tolerance, which are beneficial in industrial applications.

## Engineered microbial α-amylases with altered substrate and product specificities

Microbial α-amylases with distinct product specificity are favourable for food industry especially in starch conversion. However, the engineering of these α-amylases for altered substrate specificity was scarce. Although many SDM studies had elucidated various residues with significant roles in different hydrolytic activities of α-amylases (*Svensson, 1994*), only a very early study showed the K209R mutant of *A. oryzae* α-amylase (Taka-amylase A; TAA) exhibited reduced and increased specificities towards starch and *p*-nitrophenylmaltoside (G2-PNP) respectively, which involved the activity switch from α-amylase to maltosidase (*Nagashima et al., 1992*). Therefore, activity switch, substrate specificity and product specificity (hydrolysis pattern) are closely related, although substrate specificity is often neglected in most studies.

As aforementioned, SBS and CBM are important in starch (and other insoluble substrates) adsorptivity and catalytic activity of microbial α-amylases. Although CBMs have been fused with other carbohydrases (*Furukawa et al., 2013*; *Zhang et al., 2013*), microbial α-amylases have not been designed and verified in vitro on the fusion of CBM and/or SBS to alter their substrate specificity. Notably, while the CBMs are often linked to the catalytic domain via a polypeptide linker, *Zhang et al. (2017)* has reported the increase of substrate specificity towards soluble starch (72.8%), glycogen (69.3%), dextrin (74.4%), γ-cyclodextrin (65.3%), and raw starch (83.1%) when CBM20 was linked to the Domain C of *Talaromyces leycettanus* JCM12802 α-amylase (Amy13A) via a 21-peptide linker homologous to an acid-stable α-amylase (asAA) from *Aspergillus kawachii* (*Kaneko et al., 1996*).

Notably, a site-directed mutagenesis study by *Amalia et al. (2016)* had shown that the Y401W mutation of the α-amylase from *Saccharomycopsis fibuligera* R64 (mSfamyR64) had caused ≈10% increase in specificity towards soluble (7.1 to 7.8 U/mL) and raw (4.5 to 4.9 U/mL) starches compared to its wild type (rSfamyR64). Although such improvement of activity was proposed to be contributed by stronger interaction with the starch substrates, both the wild type and mutated SfamyR64 did not adsorb onto the insoluble raw starch, a desired characteristic portrayed by microbial α-amylases when CBM and/or SBS were present (*Amalia et al., 2016*).

To this end, the similar research group has adopted computational analysis to design and simulate the mutated SfamyR64 in silico. *Yusuf et al. (2017)* has evident the lacking of SBS in SfamyR64 and therefore mutating two hydrophilic amino acid residues to hydrophobic residues (S383Y/S386W) to introduce an SBS in the Domain C of SfamyR64. Molecular dynamic (MD) study (20 ns) and pairwise decomposition of interaction energy between maltose and SBS had proven the amelioration of the dynamics and binding affinity (86.5% increase in interaction energy) of mutant's SBS to the substrate, which made it comparable to the positive control (*Aspergillus niger* α-amylase) (*Yusuf et al., 2017*).

Recently, *Baroroh et al. (2019)* have extended the MD study up to 100 ns, observing that the maltose substrate was leaving the SBS at 20 ns for 12 Å away from the initial coordinate and surprisingly for 146 Å at 100 ns. The group has therefore developed a mutant with seven substitutions (S383Y/S386W/N421G/S278N/A281K/Q384K/K398R) and an insertion of a

four-residue flexible loop (G400_S401insTDGS) to ensure better starch adsorptivity and affinity (36.6%) as evaluated via the molecular generalized Born surface area (MM/GBSA) method (*Baroroh et al., 2019*). N421G reduced the stearic hindrance of the arginine residue towards W386 which was supposed to interact with the substrate in the positive control.

A281 and Q384 which were positioned above the SBS provided extra cavity for the substrate movement due to the smaller alanine residue and lack of extra hydrogen bonds. Hence, these two residues were replaced with lysine (A284K/Q384K) together with K398R (located below the SBS) to enhance the SBS affinity towards substrate through stronger hydrophobic interactions and hydrogen linkages. Nevertheless, the four-residue loop insertion (G400_S401insTDGS) also increased the number of hydrogen bonds, stabilizing the SBS structure in SfamyR64 (*Baroroh et al., 2019*). However, the latest in vitro engineering of microbial α-amylases on substrate affinity and specificity was reported by *Amalia et al. (2016)* and *Zhang et al. (2017)*. Therefore, such engineering deserves more attention from the researchers in the future.

A site-saturation mutagenesis study (*Li et al., 2018*) had revealed on the higher affinity of ROAmy mutants H286L and H286M towards both maltotriose (G3) and soluble starch, which were correlated with the high-level production of maltose as end-products. Based on the molecular docking analysis, such phenomenon was attributed to the new non-polar contacts formed between the hydrophobic residues (leucine and methionine) with G3 at the non-reducing end (subsite −1) whereby extra hydrogen bonds could be formed between two more amino acid residues (*Li et al., 2018*).

In addition, maltohexaose (G6) specificity was also enhanced in the Bst-MFAse mutants (G109N, G109D, G109F) with G6 production from starch at 36.1, 42.4 and 39.0% respectively compared to only 32.9% in wild type (*Xie et al., 2020*). Through homology modelling analysis, the group (*Xie et al., 2020*) discovered that extra interactions (hydrogen bonds or hydrophobic interaction) were formed between the replaced residues and the substrate at subsite -6, promoting the hydrolysis pattern from starch to G6 by the MFAse (Fig. 3). On the contrary, a previous study by the group presented that the Bst-MFAse mutants (W139A, W139L, W139Y) exhibited significant increase in maltopentaose (G5) production and aglycone-productive binding but had counter effect on maltohexaose (*Xie et al., 2019b*). It was suggested by the authors that the aromatic stacking between W139 and the substrate could control Bst-MFAse's product specificity and its oligosaccharide hydrolysis pattern.

Besides, a rational engineering of *B. subtilis* CN7 α-amylase (*Wang et al., 2020*) was performed to generate 5 mutants (Y204F, Y204I, Y204V, V260I, V260L) with altered product specificities. In the study, all created mutants were found to produce more glucose (G1) than maltose (G2), with Y204V as exception. Interestingly, Y204F exhibited no G3 production, while Y204V produced G2, G3 and G5 without releasing any G1 and maltotetraose (G4) (*Wang et al., 2020*). The abolished hydrolysis to produce G1 as the final hydrolytic product had convert this typical α-amylase to a novel MFAse which was preferred in bread and high-fructose corn syrup (HFCS) industries (*Pan et al., 2017*). However, it is worth mentioning that both Y204 and V260 were present in the β-strands of central TIM-barrel which contributed to the α-amylase catalytic ability.
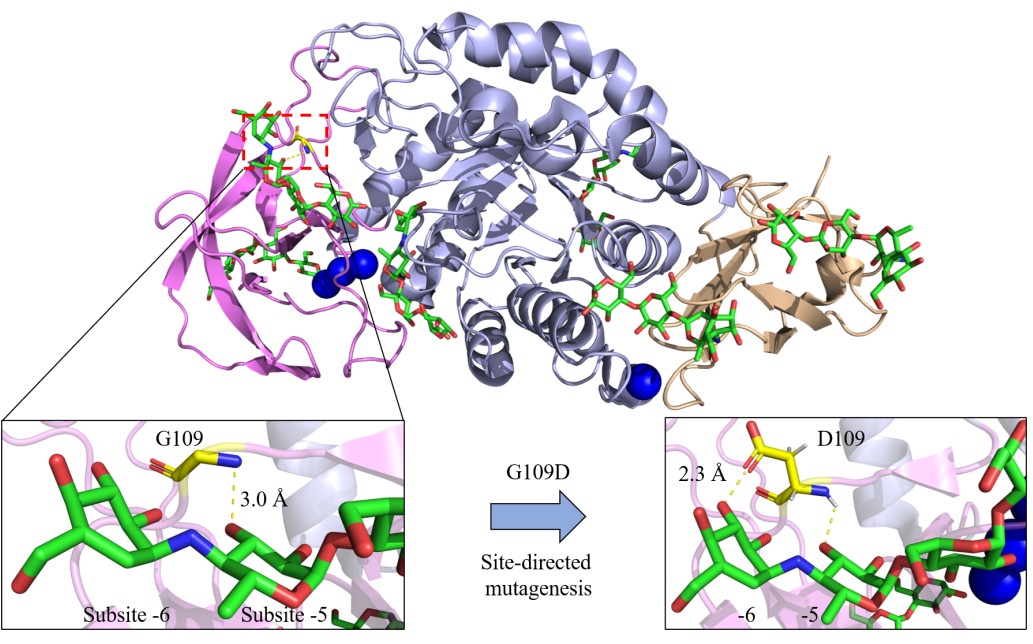

**Figure 3** **Site-directed mutagenesis of G109D in Bst-MFAse** (**PDB ID: 6ag0;** *Xie et al., 2019a*). Extra interaction at subsite -6 has been established with the introduction of D109, leading to the highest increment of the G6 production from starch at 42.4% compared to other mutants (G109N and G109F) and its wild type (G109) (*Xie et al., 2020*). The structural figures were illustrated using the PyMOL Molecular Graphic System, Version 2.4, Schrödinger, LLC.

While some α-amylases also possess transglycosylation activity, such activity is responsible for G3 formation which is undesired for high maltose syrups industry (*Mehta & Satyanarayana, 2016*). To enhance the ratio of hydrolysis to transglycosylation in α-amylases, several strategies encompassing the control of water accessibility, reduction of steric interference, improvement of acceptor molecule binding and modulation of catalytic nucleophile $pK_a$ could be performed (*Abdul Manas, Md. Illias & Mahadi, 2018*). One such example is the SDM of acceptor binding subsite +2 in *B. stearothermophilus* maltogenic amylase by replacing W177 with phenylalanine (F), tyrosine (Y), leucine (L), asparagine (N) and serine (S), which reported on the reverse-proportionate relationship between transglycosylation activity and the hydrophilicities of the replaced residues (*Sun et al., 2016*). With the increased hydrophilicity at subsite +2, water molecule was more readily bound to promote G2 production through hydrolysis, whereby W177S mutant was most promising and superior for industrial application (*Sun et al., 2016*).

**Engineered microbial α-amylases with increased oxidative stability**
Oxidative stability is an important yet highly preferred property for the α-amylases to be utilized in detergent industry. Cysteine and methionine residues which had sulfhydryl group (R-SH) had been determined as two most oxidation prone residues in α-amylases (*Brosnan, Kelly & Fogarty, 1992*). While most cysteine residues form disulphide bridge to increase α-amylase's stability, more studies had been performed to mutate methionine to other oxidative-resistant residues as first exemplified in *B. licheniformis* α-amylase (BLA)

(*Brzozowski et al., 2000*) and *Bacillus* sp. strain KSM-1378 α-amylase (AmyK) (*Igarashi, Hagihara & Ito, 2003*).

An SDM study (M197A) had generated a *G. stearothermophilus* US100 α-amylase mutant (AmyUS100/ ΔI214-G215, M197A) with 70% residual activity after 60 min of 1.8 M hydrogen peroxide ($H_2O_2$) treatment at 60 °C (*Khemakhem et al., 2009*). Similarly, the corresponding methionine residue of AmyUS100 in the truncated *Bacillus* sp. strain TS-23 (BAC ΔNC) was mutated (M231L), resulting in mutants which retained more than 96% original activity in the presence of 500 mM $H_2O_2$ (*Chi et al., 2010*). Nevertheless, a structural-based SDM study (M145L, M214L, M229L, M247L, M317L) of *Alkalimonas amylolytica* α-amylase had shown the mutations of these near-to-active-site methionine residues had enhanced oxidative stability, with M247L mutant exhibiting the highest resistance (72%) (*Yang et al., 2012*).

The subsequent research by Chen's group (*Yang et al., 2013a*) had generated 85 multiple mutants based on 8 single mutants with enhanced oxidative stability, where an outstanding multiple mutant (M145I-214A-229T-247T-317I) had exhibited 5.4-fold improvement in oxidative stability at 91.3% of original activity when incubated with 500 mM $H_2O_2$ for 1 h. Interestingly, the *A. amylolytica* α-amylase with its N-terminal fused with an highly hydrophilic oligopeptide which formed β-sheet structure in aqueous solution had 2.7-fold increase in oxidative resistance while retaining 54% of its original activity after 30 min incubation with 500 mM $H_2O_2$ (*Yang et al., 2013b*). Such observation was justified with the further distance of M247 with the catalytic residues (D278 and D340) in the mutant compared to its wild type, as previously exemplified (*Yang et al., 2012*; *Yang et al., 2013b*).

The latest investigation on engineering of α-amylases for enhanced oxidative stability, was perhaps in 2013. However, this study (*Ozturk et al., 2013*) had proven that the mutation at near-to-active-site methionine residue (M55A) was more significant than mutation of solvent-accessible methionine residues (M43A + M44A) in terms of oxidative stability, where both types of methionine residues were proven to be oxidation prone (*Lin et al., 2003*; *Yang et al., 2012*). This statement was justified when M55A mutant retained 50% of its initial activity in the presence of 100 mM $H_2O_2$, compared to double mutant (M43A + M44A) at 39% (*Ozturk et al., 2013*). Nevertheless, such investigations and engineering to improve oxidative stability of microbial α-amylases remain underexplored and should not become obsolete (latest report in 2013; Table 2) due to its great interests and advantages in detergent industry.

## DISCUSSIONS

Microbial α-amylases remain as valuable assets in the industries for their favourable properties encompassing thermostability, pH stability and tolerance, product and substrate specificities as well as oxidative stability. These native and recombinant α-amylases can be produced intracellularly or extracellularly, purified and further characterized for their important biochemical properties. Although there have been a wide variety of purification techniques available, microbial α-amylases are often purified at higher recovery and purification fold using affinity chromatography when they are expressed recombinantly with the frequently fused polyhistidine tags (*Lim, Oslan & Oslan, 2020*).

The purified microbial α-amylases can be extensively characterized for industrially favoured traits. These characteristics are mutually reflective to the 3D structural properties. Albeit the classical three-domain fold of most microbial α-amylases, the presence of CMB (*Armenta et al., 2017*) and/or SBS (*Baroroh et al., 2017*) is crucial for their substrate and product specificities. The interactions within the amino acid residues in the enzymes, encompassing salt bridges, disulfide linkages, hydrogen bonds and hydrophobic interactions are undeniably attributed to their biochemical properties, especially the temperature and pH optimal, as well as the stability towards extreme conditions.

Although these microbial α-amylases might initially exhibit the biochemical traits which are preferred for industrial applications, the more powerful enzymes have continuously been sought. Greater abilities to withstand the harsh environments are advantageous for more specifically targeting or widening of their usages in a single or several different industries, respectively. In brief, enzymes having higher optimum temperature and thermostability as well as adequate substrate and product specificities are favourable in the food and beverages industries. In detergent industry, however, resistance to extreme pH and oxidation is crucial for the enzymes to remain functional when they are added as the detergent additives. While the protein engineering techniques have been more advanced, the frequently used strategies to modify microbial α-amylases are rational design, directed evolution, truncation and terminal fusion (*Sharma et al., 2021*).

Rational design is an *in silico* approach to determine the residue(s) or region(s) to be mutated, which is generally based on an MSA within gene sequences of the congeneric species (*Yang et al., 2017*). Besides MSA, rational design can also be a systematic structure-based strategy (*Yang et al., 2013a*) which is a targeted engineering compared to directed evolution (error prone PCR and DNA shuffling). Directed evolution is essentially a labour-intensive and time-consuming approach to screen all the mutants for the improved and desired biochemical properties. The engineering works reviewed in this article have been evident to enhance the biochemical properties of microbial α-amylases encompassing longer half-life at extreme temperature (100 °C), pH (pH 3.5 and 10.0) and oxidative stress (1.5 M $H_2O_2$) as well as enhanced specificities on substrates (>65.3%) and products (42.4%). Therefore, these ameliorations of microbial α-amylases characteristics have successfully proven that the protein engineering techniques are indeed promising and worth exploring to generate new yet valuable mutants for better suitability in various industrial applications.

## CONCLUSIONS

To conclude, the engineering of microbial α-amylases should be a continuous and sustained effort to design the enzymes for their pertinent roles in various industrial applications. Therefore, a review covering the current protein engineering techniques and design rationales is needed to provide conceptual advances to the researchers. The critical points of mutations are highly dependent on the respective α-amylases. Disulfide bridge establishments by converting polar amino acids into cysteines is observed for enhanced thermostability. Residues substituted into basic amino acids (lysine, histidine and arginine)

are remarkable for acidic resistance. Hydrophobic residues for hydrophobic platform in the CBM/SBS are preferred for improved substrate specificities. Oxidation-prone methionine residues are often substituted with non-polar residues (alanine, leucine, isoleucine and phenylalanine) to enhance its oxidative stability. Notably, the engineering investigations on ameliorated substrate specificity and oxidative stability are limited and deserve further exploration. It could be conducted as previously performed but inferable from other biocatalysts (lipases, proteases and others). Cross-disciplinary collaborations among researchers having the expertise of structural biology, metabolism and fermentation can be established to produce microbial α-amylases with enhanced properties in industrial-scale bioreactors. Thus, microbial α-amylases with various preferred characteristics are possible with careful scrutinization of their sequential, structural and evolutionary analyses.

## ACKNOWLEDGEMENTS

Sincere thanks to all members of Enzyme and Microbial Technology (EMTech) Research Centre and Enzyme Technology Laboratory, VacBio 5, Institute of Bioscience, Universiti Putra Malaysia.

### Funding

This work was supported by Universiti Putra Malaysia (Putra-IPS grant number GP-IPS/2016/9513300). The funders had no role in study design, data collection and analysis, decision to publish, or preparation of the manuscript.

### Grant Disclosures

The following grant information was disclosed by the authors:
Universiti Putra Malaysia: GP-IPS/2016/9513300.

### Competing Interests

The authors declare there are no competing interests.

### Author Contributions

- Si Jie Lim conceived and designed the experiments, performed the experiments, analyzed the data, prepared figures and/or tables, authored or reviewed drafts of the paper, and approved the final draft.
- Siti Nurbaya Oslan conceived and designed the experiments, performed the experiments, authored or reviewed drafts of the paper, and approved the final draft.

### Data Availability

There are no raw data or code used in this literature review.

### Supplemental Information

Supplemental information for this article can be found online at http://dx.doi.org/10.7717/peerj.11315#supplemental-information.

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
