# Peer review of "Native to designed: microbial -amylases for industrial applications"

_PeerJ, doi:10.7717/peerj.11315_

## Round 0.1 · original submission · Major Revisions

An interesting review article, as also highlighted by reviewers. However, it needs some corrections for better understanding and clarity. Kindly go through their suggestions, and revise your manuscript accordingly.

·

Basic reporting

The Manuscript is written well.
The structure conforms to PeerJ standards

Experimental design

Article content is within the Aims and
Scope of the journal.
Yes
Rigorous investigation performed to a
high technical & ethical standard.
yes
Methods described with sufficient detail &
information to replicate.
NO
Is the Survey Methodology consistent with
comprehensive, unbiased coverage of
the subject? If not, what is missing?
No, the difference between alpha and beta-amylase is missing, the mode of action is missing.
Are sources adequately cited? Quoted or
paraphrased as appropriate?
Yes
Is the review organized logically into
coherent paragraphs/subsections?
No

Validity of the findings

Impact and novelty not assessed. Negative/inconclusive results accepted. Meaningful replication encouraged where rationale & benefit to literature is clearly stated.
Yes
Conclusions are well stated, linked to original research question & limited to supporting results.
No

Is there a well developed and supported argument that meets the goals set out in the Introduction?
No

Does the Conclusion identify unresolved questions / gaps / future directions?
No

Additional comments

The Manuscript is written well, but there are some comments in the body of the manuscript that need to be improved.
The authors must explain the mode of action of a-amylase.
Explain what the difference between alpha and beta-amylase in the mechanism of catalysis.
The abstract is descriptive and qualitative. Normally an abstract should state briefly the purpose of the study undertaken and meaningful conclusions based on the obtained results. Hence, this needs rewriting. I would expect a brief, yet concise, quantitative data description of the results in the abstract.
The conclusion should contain more specific values to reflect the significance of the work.

Reviewer 2 ·

Basic reporting

No comment

Experimental design

No comment

Validity of the findings

No comment

Additional comments

On the whole, the manuscript reviews the reports concerning producing native or engineered microbial α amylases for application quite thoroughly, thus it will be a good guideline literature for the researchers of the area. However, before publication, several issues listed below are suggested to be addressed.

1. In the most situations in the paper, “engineering” seems to denote “changing amino acid sequence by mutation including substitution, truncation, insertion or fusion”, then “…engineering and mutations…” in the line 74 should be corrected to avoid confusion.
2. Since the paper includes many examples of producing native or recombinant α amylases by homologous or heterologous expression without changing the amino acid sequences (engineering), it would better be titled “Native to designed: microbial α amylases for industrial applications”.
3. Although the authors state the desirable traits of α amylase for every specific applications, it would be better to summarize all the desirable traits and corresponding applications in a table for clearness.
4. In section “Structure Properties if Microbial α amylase”, another figure that diagrams the general architecture of α amylases including domain A, B, C and CBM/SBD would be better for readers understanding their common features. In addition, more detailed structural characteristics such as hydrophobic residues on the surface of SBS and SBD should be provided in the text to help the readers to understand the “rationale” of “rational design”.
5. Figures showing the specifically engineered structure are favorable for understanding the examples in “Engineered microbial α Amylase with Altered Substrate and Product Specificities”.

Reviewer 3 ·

Basic reporting

The paper is well designed. However, it does not take into account cold-adapted alpha-Amylases and, most importantly, from protists. These enzymes have high specific activity at low and moderate temperatures, a property that can be extremely useful in various industrial applications as it implies a reduction in energy consumption during the catalyzed reaction.
Therefore, I suggest adding to the review some results reported in the two following papers:
1: Yang G, Yao H, Mozzicafreddo M, Ballarini P, Pucciarelli S, Miceli C.
Rational Engineering of a Cold-Adapted α-Amylase from the Antarctic Ciliate
Euplotes focardii for Simultaneous Improvement of Thermostability and Catalytic
Activity. Appl Environ Microbiol. 2017 Jun 16;83(13):e00449-17. doi:
10.1128/AEM.00449-17. PMID: 28455329; PMCID: PMC5478988.

2: Yang G, Yang G, Aprile L, Turturo V, Pucciarelli S, Pucciarelli S, Miceli C.
Characterization and comparative analysis of psychrophilic and mesophilic alpha-
amylases from Euplotes species: a contribution to the understanding of enzyme
thermal adaptation. Biochem Biophys Res Commun. 2013 Sep 6;438(4):715-20. doi:
10.1016/j.bbrc.2013.07.113. Epub 2013 Aug 2. PMID: 23916704.

which report the comparison of alpha-amylases from two congeneric species adapted to two different environments. This comparison allows identifying residues to be changed with rational design by site-directed mutagenesis.

I suggest adding more figures that represent the important sites of the alpha-amylases.

In my opinion, the paper is acceptable after these minor modifications.

Experimental design

The study is well designed. However, I would check better the results obtained from the bibliography databases. For example, I would add extremophiles, microbial and cold stability.

Validity of the findings

The findings are well described. In my opinion, there is a lack of more information on the structure and amino acid sequence of the alpha-amylase.

Additional comments

The paper is well designed. However, it does not take into account cold-adapted alpha-Amylases and, most importantly, from protists. These enzymes have high specific activity at low and moderate temperatures, a property that can be extremely useful in various industrial applications as it implies a reduction in energy consumption during the catalyzed reaction.
Therefore, I suggest adding to the review some results reported in the two following papers:
1: Yang G, Yao H, Mozzicafreddo M, Ballarini P, Pucciarelli S, Miceli C.
Rational Engineering of a Cold-Adapted α-Amylase from the Antarctic Ciliate
Euplotes focardii for Simultaneous Improvement of Thermostability and Catalytic
Activity. Appl Environ Microbiol. 2017 Jun 16;83(13):e00449-17. doi:
10.1128/AEM.00449-17. PMID: 28455329; PMCID: PMC5478988.

2: Yang G, Yang G, Aprile L, Turturo V, Pucciarelli S, Pucciarelli S, Miceli C.
Characterization and comparative analysis of psychrophilic and mesophilic alpha-
amylases from Euplotes species: a contribution to the understanding of enzyme
thermal adaptation. Biochem Biophys Res Commun. 2013 Sep 6;438(4):715-20. doi:
10.1016/j.bbrc.2013.07.113. Epub 2013 Aug 2. PMID: 23916704.

which report the comparison of alpha-amylases from two congeneric species adapted to two different environments. This comparison allows identifying residues to be changed with rational design by site-directed mutagenesis.

I suggest adding more figures that represent the important sites of the alpha-amylases. There is a lack of more information on the structure and amino acid sequence of the alpha-amylase.

Furthermore, I would check better the results obtained from the bibliography databases. For example, I would add extremophiles, microbial and cold stability.

---

## Round 0.2 · accepted · Accept

The authors revised the manuscript considering all suggestions and it can be accepted in its current version.

·

Basic reporting

acceptable

Experimental design

acceptable

Validity of the findings

acceptable

Additional comments

The authors made all modifications and therefore I recommended that the manuscript be accepted as it is

Reviewer 2 ·

Basic reporting

No comment.

Experimental design

No comment.

Validity of the findings

No comment.

Additional comments

No comment.

Reviewer 3 ·

Basic reporting

The paper has been improved. The authors replied to all reviewer's comments. The paper now is ready for acceptance.

Experimental design

The study design has been improved. The authors added more information about the topic and the methodology.

Validity of the findings

Now the findings are real valid and supported.

Additional comments

The paper can be accepted in this form.